# AIRV2X: UNIFIED AIR-GROUND VEHICLE-TO-EVERYTHING COLLABORATION

## ABSTRACT

While multi-vehicular collaborative driving demonstrates clear advantages over single-vehicle autonomy, traditional infrastructure-based V2X systems remain constrained by substantial deployment costs and the creation of "uncovered danger zones" in rural and suburban areas. We present AirV2X-Perception, a large-scale dataset that leverages Unmanned Aerial Vehicles (UAVs) as a flexible alternative or complement to fixed Road-Side Units (RSUs). Drones offer unique advantages over ground-based perception: complementary bird's-eye-views that reduce occlusions, dynamic positioning capabilities that enable hovering, patrolling, and escorting navigation rules, and significantly lower deployment costs compared to fixed infrastructure. Our dataset comprises 6.73 hours of drone-assisted driving scenarios across urban, suburban, and rural environments with varied weather and lighting conditions. The AirV2X-Perception dataset facilitates the development and standardized evaluation of Vehicle-to-Drone (V2D) algorithms, addressing a critical gap in the rapidly expanding field of aerial-assisted autonomous driving systems. The dataset and development kits are open-sourced at https://anonymous.4open.science/r/AirV2X-Perception-BBA7.

## 1 INTRODUCTION

Multi-vehicular collaborative driving has been shown to be more effective than single-vehicle autonomous driving primarily due to the complementary sensory coverage provided through inter-vehicle cooperation. This collaborative approach significantly reduces perception limitations (Zhang et al., 2023; Wang et al., 2025d; Gao et al., 2024) by enabling vehicles to share sensor data and compensate for each other's blind spots. Recent research in Vehicle-to-Everything (V2X) communication demonstrates that Road-Side Units (RSUs) mounted on infrastructure offer superior perception capabilities than vehicle-only collaboration, primarily due to their elevated positioning that minimizes blind spots and provides broader field-of-view coverage. However, traditional infrastructure-based solutions face significant economic constraints. The substantial construction and maintenance costs associated with RSUs necessitate strategic deployment decisions. Consequently, these units are predominantly installed at high-traffic intersections and critical urban junctions to maximize cost-benefit ratios. This economically-driven compromise results in what we term "uncovered danger zones"—highways, pedestrian zones, suburban neighborhoods, and rural areas where vehicles operate without the perceptual advantages of infrastructure support.

**Motivation.** In parallel with these infrastructure limitations, the rapid rise of the low-altitude economy and steady advances in Unmanned Aerial Vehicle (UAV) technology offer a promising alternative. Modern drones now handle everything from emergency response and fire rescue to goods transport and everyday food delivery. The evolution of drone capabilities has naturally paved the way for drone-based perception systems, which represent a logical extension of this technology into the autonomous driving domain. Unlike static infrastructure-dependent solutions, drone-assisted collaborative driving leverages the mobility and versatility of aerial platforms to enhance vehicular perception in environments where traditional RSUs are economically unfeasible. Compared to traditional RSUs, drone-based perception systems offer several significant advantages:

- **Drones offer a unique and complementary perception perspective** distinct from both vehicle-mounted sensors and fixed RSUs. Ground vehicles capture primarily horizontal, ego-centric views limited by occlusions, while RSUs provide fixed vantage points typically restricted to intersections. In contrast, drones deliver real-time bird's-eye views that are both elevated and mobile,

enabling holistic scene understanding in cluttered environments. This aerial perspective facilitates tracking of agents across occlusions, better anticipation of multi-agent interactions, and improved detection of lane changes or cut-ins otherwise hidden from ground-level sensors.

- **Drones provide superior operational adaptability and flexibility** through dynamic positioning based on real-time requirements. They can hover above specific areas of interest, patrol designated routes, or be programmed to escort individual vehicles, delivering tailored perception assistance precisely where and when needed—capabilities that fixed infrastructure simply cannot match.

- **Drone solutions present a more cost-effective approach to expanding coverage**. According to the 5G Automotive Association (5GAA) in 2020 (Nokes et al., 2020), a fixed RSU without camera or lidar sensors costs between $20,000-80,000$. In comparison, fully-equipped drones with high-resolution cameras and lidar range from under $1,000$ to $50,000$ depending on specifications. This economic flexibility enables strategic resource allocation—deploying less expensive Vehicle-to-Drone (V2D) systems in rural areas while positioning advanced systems in accident-prone or high-density zones.

- **V2D systems can be integrated with emerging aerial networks** as the low-altitude economy develops. The increasing presence of UAVs performing various tasks creates opportunities for "opportunistic" sensing and communication channels for ground vehicles. These aerial vehicles, already equipped with advanced sensors and communication devices, can establish a multi-layered perception network with minimal additional infrastructure investment, creating a more robust and redundant system for autonomous vehicle perception.

These compelling advantages have sparked considerable interest in Vehicle-to-Drone (V2D) communication systems, leading to substantial research efforts focused on developing algorithms that can effectively utilize aerial perspectives to enhance ground vehicle autonomous driving. As a result, there has been a significant increase in research exploring various aspects of drone-assisted vehicular perception, from communication protocols to collaborative sensing frameworks (Hu et al., 2022; Gao et al., 2025b;a; Wu et al., 2025a). However, with the flourishing of V2D algorithms and the demand for more advanced solutions, there remains a critical absence of high-quality datasets specifically designed for training and evaluating drone-assisted perception systems. This limitation hampers the development of robust algorithms that can effectively leverage aerial perspectives to enhance vehicular perception.

To bridge this gap, we present **AirV2X-Perception**, **a large-scale drone-assisted V2X driving dataset**. AirV2X-Perception contains 6.73 hours of drone-assisted driving data collected in the co-simulation of the CARLA (Dosovitskiy et al., 2017) and Airsim (Shah et al., 2018) simulator. The dataset contains multiple connected agent types including vehicles, roadside units (RSU), and drones, each equipped with different sensors. The dataset is collected in various urban and rural areas, with different weather (clear, rainy, foggy, and cloudy) and lighting conditions (daytime, dusk, and nighttime). AirV2X-Perception also contains various common navigation strategies for drones, including hovering, patrolling, and escorting, to provide a more comprehensive evaluation of the V2D algorithms. To accommodate the challenges of large-scale perception networks, we include up to 15 connected agents simultaneously (5 vehicles, 5 RSUs, 5 drones) in single scenarios. By releasing this comprehensive dataset to the research community, we aim to accelerate the development of robust drone-assisted perception algorithms and establish standardized benchmarks for performance evaluation.

## 2 RELATED WORKS

### 2.1 V2X DATASETS

In this section, we summarize some widely used V2X datasets in Table 1. Existing V2X datasets (Wang et al., 2025b) can be categorized into three distinct groups. The **Vehicle + Infrastructure** category constitutes the majority of current datasets, with simulated environments like OPV2V (Xu et al., 2022d), V2XSim (Li et al., 2022), and V2XSet (Xu et al., 2022c) providing multiple connected vehicles and infrastructure elements for various perception tasks. Real-world counterparts such as DAIR-V2X (Yu et al., 2022), TUMTraf (Zimmer et al., 2024), and V2XReal (Xiang et al., 2024) offer authentic data but typically suffer from limited scale and environmental diversity, predominantly focusing on urban daytime scenarios. The **Drone-specific** category includes datasets such as CoPercpetion-UAV (Hu et al., 2022), CoPercpetion-UAV+ (Hu et al., 2023), UAV3D (Ye

Table 1: '–' denotes an unavailable attribute. 'R/S' refers to whether the data is real-world or simulated. 'A.W.' denotes adverse weather conditions. 'D/N' indicates daytime or nighttime scenarios, while 'U/R' refers to urban or rural environments.

| Year | Dataset | R/S | Max Connected Agents | | | Modalities | | Diversity | | |
|---|---|---|---|---|---|---|---|---|---|---|
| | | | CAVs | Infra | Drones | Cam. | LiDAR | A.W. | D/N | U/R |
| *Vehicle + Infrastructure* | | | | | | | | | | |
| Xu et al. (2022d) | OPV2V | Sim | 7 | 0 | 0 | ✓ | ✓ | – | D | U |
| Li et al. (2022) | V2XSim | Sim | 5 | 1 | 0 | ✓ | ✓ | – | D | U |
| Xu et al. (2022c) | V2XSet | Sim | 7 | 1 | 0 | ✓ | ✓ | – | D | U |
| Yu et al. (2022) | DAIR-V2X | Real | 2 | 1 | 0 | ✓ | ✓ | – | D+N | U |
| Xu et al. (2023) | V2V4Real | Real | 2 | 0 | 0 | – | ✓ | – | D | U |
| Hao et al. (2024) | Rcooper | Real | 0 | 4 | 0 | ✓ | ✓ | – | D+N | U |
| Zimmer et al. (2024) | TUMTraf | Real | 1 | 1 | 0 | ✓ | ✓ | ✓ | D | U |
| Ma et al. (2024) | HoloVIC | Real | 1 | 4 | 0 | ✓ | ✓ | – | D | U |
| Xiang et al. (2024) | V2XReal | Real | 4 | 2 | 0 | ✓ | ✓ | – | D | U |
| Huang et al. (2024) | V2X-Radar | Real | 1 | 1 | 0 | ✓ | ✓ | ✓ | D+N | U |
| *Drones Only* | | | | | | | | | | |
| Hu et al. (2022) | CoP-UAV | Sim | 0 | 0 | 5 | ✓ | – | – | D | U |
| Hu et al. (2023) | CoP-UAV+ | Sim | 0 | 0 | 10 | ✓ | – | – | D | U |
| Ye et al. (2024) | UAV3D | Sim | 0 | 0 | 5 | ✓ | – | – | D | U |
| Feng et al. (2024) | U2Udata | Sim | 0 | 0 | 3 | ✓ | ✓ | ✓ | D+N | R |
| *Vehicle + Infrastructure + Drones* | | | | | | | | | | |
| Dutta et al. (2024) | MAVREC | Real | 1 | 0 | 1 | ✓ | – | – | D | U+R |
| Wang et al. (2024) | UVCPNet | Sim | 1 | 0 | 2 | ✓ | – | – | D | U |
| Wang et al. (2025a) | Griffin | Sim | 1 | 0 | 1 | ✓ | ✓ | ✓ | D+N | U+R |
| Hou et al. (2025) | AGC Drive | Real | 2 | 0 | 1 | ✓ | ✓ | – | D+N | U+R |
| – | AirV2X (Ours) | Sim | 5 | 5 | 5 | ✓ | ✓ | ✓ | D+N | U+R |

et al., 2024), and U2Udata (Feng et al., 2024), which concentrate on aerial vehicle collaboration but lack the ground-vehicle components essential for comprehensive V2X research. The emerging **Vehicle + Infrastructure + Drone** category attempts to integrate all three agent types, but current offerings like MAVREC (Dutta et al., 2024), UVCPNet (Wang et al., 2024), Griffin (Wang et al., 2025a), and AGC Drive (Hou et al., 2025) exhibit significant limitations—restricted to minimal agent configurations (typically one vehicle with one or two drones), supporting limited perception tasks, and lacking environmental diversity. Our proposed AirV2X-Perception dataset addresses these limitations by providing an unprecedented comprehensive solution integrating all three agent types at scale. With support for 5 CAVs, 5 infrastructure elements, and 5 drones, it offers the most extensive connected agent environment currently available. Unlike existing datasets with limited scenarios, AirV2X-Perception encompasses diverse environmental conditions spanning urban and rural settings, daytime and nighttime operations, and various adverse weather conditions. Furthermore, it supports both camera and LiDAR modalities across vehicular and RSU agent types alongside high-resolution camera sensors for drone agents, enabling research on multi-modal and cross-modal perception algorithms. The dataset facilitates various perception tasks including object detection, semantic segmentation, and tracking, making it versatile for different collaborative perception research directions while providing a realistic testbed for evaluating algorithms under conditions closer to real-world deployment scenarios.

## 2.2 V2X PERCEPTION ALGORITHMS

**Fusion scheme taxonomy** is a primary categorization framework for V2X collaborative perception algorithms. **Early fusion** (Gao et al., 2018; Chen et al., 2019b; Arnold et al., 2020) involve direct sharing of raw sensor data, maximally preserving information but requiring prohibitive bandwidth for practical deployment. **Late fusion** methods (Melotti et al., 2020; Fu et al., 2020; Zeng et al., 2020; Shi et al., 2022; Glaser & Kira, 2023) share only final predictions, dramatically reducing communication overhead at the cost of suboptimal accuracy due to information loss. **Intermediate fusion** techniques (Wang et al., 2020; Liu et al., 2020; Cui et al., 2022; Xu et al., 2022b; Qiao & Zulkernine, 2023; Li et al., 2023; Wang et al., 2023; Yu et al., 2023; Wang et al., 2025c) represent the most widely adopted approach, striking a balance by sharing mid-level representations (e.g., BEV features) that enable flexible collaboration while maintaining reasonable data transmis-

sion bandwidth. The emerging **language-based fusion** paradigm (Luo et al., 2025a; You et al., 2024; Wu et al., 2025b; Gao et al., 2025a; Luo et al., 2025b) offers advantages in transmission efficiency, explainability, and interoperability, though our work focuses primarily on benchmarking the predominant intermediate fusion approaches.

**Key technical challenges** in V2X collaborative perception include model complexity, communication efficiency, and agent heterogeneity. Early transformer-based architectures like V2X-ViT (Xu et al., 2022c) and CoBEVT (Xu et al., 2022a) achieved significant performance improvements over predecessors such as F-Cooper (Chen et al., 2019a) and V2VNet (Wang et al., 2020), but at the cost of **high computational demands**. Subsequent work has addressed these limitations through various strategies: SICP (Qu et al., 2024) employs convolutional neural networks to reduce model complexity, while HEAL (Lu et al., 2024) and STAMP (Gao et al., 2025b) implement hierarchical fusion strategies that enhance feature processing while scaling to larger agent numbers. For **communication efficiency**, When2com (Liu et al., 2020) introduced selective communication through graph grouping, and Where2comm (Hu et al., 2022) utilized spatial confidence maps to share only partial feature maps—significantly reducing bandwidth requirements while preserving accuracy. **Agent heterogeneity** presents another challenge in V2X systems. HEAL (Lu et al., 2024) addressed this through backward alignment strategies. STAMP (Gao et al., 2025b) developed lightweight adapter-reverter pairs for feature alignment without modifying the local models. LangCoop (Gao et al., 2025a) innovated using natural language as a universal communication medium between diverse agents. Our AirV2X-Perception dataset features 10+ connected agents per scene and introduces the novel heterogeneity of aerial perspectives. It serves as a comprehensive benchmark for evaluating these algorithms against multiple real-world challenges, particularly in the underexplored domain of drone-assisted V2X systems.

## 3    AIRV2X-PERCEPTION DATASET

In this section, we introduce AirV2X-Perception, a novel dataset designed specifically to advance drone-assisted V2X collaborative perception research. We first detail our simulation environment and scenario design (§3.1), followed by an exploration of three distinct drone navigation strategies—hover, patrol, and escort (§3.2). We then describe our data collection methodology (§3.3) and conclude with our annotation approach and downstream task formulations (§3.4), designed to facilitate benchmark evaluations for this emerging research domain.

### 3.1    SIMULATOR ENVIRONMENT AND SCENARIO DESIGN

**Simulator Environment:** The AirV2X-Perception dataset is collected by co-simulating CARLA (Dosovitskiy et al., 2017) and Airsim (Shah et al., 2018) simulator environments. CARLA simulator is a high-fidelity open-source simulator for autonomous driving research, which provides realistic physical dynamics, vehicle control and interactions as well as photo-realistic digital assets for sensor data collection. Airsim is designed for unmanned aerial vehicle (UAV) simulation and provides realistic physical dynamics of drones. In this project, we use it to simulate the dynamics of drone agents and synchronized with actors in the CARLA simulator. We designed a total of 6.73-hour V2X collaborative driving sequences collected through Towns 1-4 and 6-7, and 12 of the CARLA simulator. We ignore Towns 5 and 10 since these maps contain static objects embedded into the CARLA static map asset that do not have accurate ground truth labels or sensing results, which may potentially lead to unstable model training. We also ignore town 11 as it is an unadorned map that does not align with the collaborative perception objectives of this work.

**Scenario Design:** The dataset is collected in four different weather conditions including clear, cloudy, foggy, and rainy, as well as three daytime type variations including day, dusk, and night, forming a variety of lighting conditions such as cloudy day, rainy night, and clear dusk, etc., assisting model training and evaluation in different weather and lighting conditions. Since the CARLA simulator provides urban and rural maps, we also collect data in both environments. The statistical distribution of the scenes in the dataset is shown in Figure 1.

### 3.2    NAVIGATION STRATEGIES FOR DRONES

We designed three types of navigation strategies for drones, hover – letting a drone hover at a fixed position, patrol – assigning each drone a list of predefined waypoints to navigate, and escort –

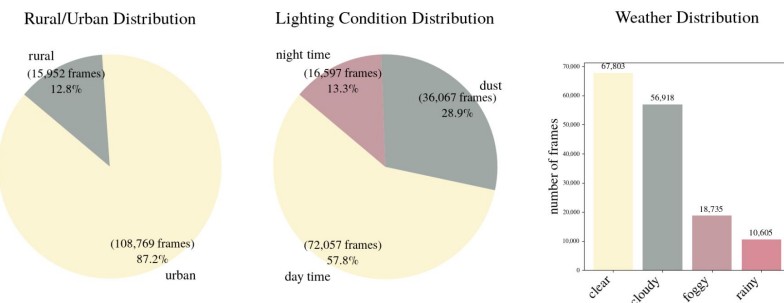

Figure 1: The percentage of each scenario in the dataset.

assigning each drone a connected vehicle to follow. Each of these navigation strategy designs has its own advantages and application in the real world. More specifically, hover is suitable for single-point monitoring in a specific area with minimum power consumption. Secondly, patrol enables the drone to cover a larger area, which is more suitable for large and complex transportation scenarios, but requires more power consumption and high complexity in multi-drone route planning. Finally, escort strategy is designed to assist a certain vehicle or a certain platooning group of vehicles. A conceptual visualization of these three trajectories is shown in Figure 2. The dataset incorporates all of these three types of trajectories. The distribution of each type of navigation strategy in the dataset is shown in Figure 2.

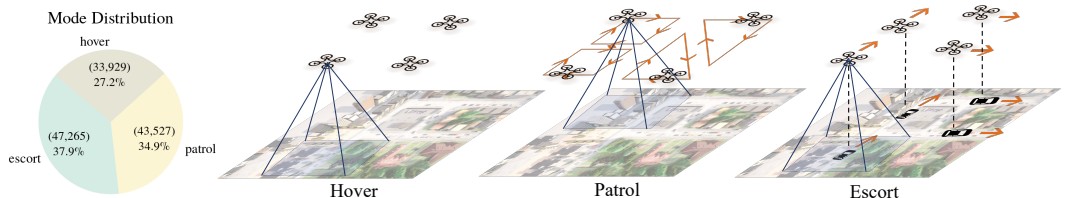

Figure 2: A conceptual visualization of three different navigation strategies for drones, along with the data distribution of each strategy.

## 3.3 DATA COLLECTION

Sensor data are collected at a 5 Hz frequency. Each driving sequence contains 3 to 5 connected vehicles, 3 to 5 connected drones, and 3 to 5 RSUs. Each agent is equipped with both LiDAR and camera sensors and the details of the sensor setups are summarized in Table 2. Each vehicle is equipped with 6 cameras forming a surround-view matrix. The camera sensor placement can be visualized in Figure 3 (c). Each agent also carries GNSS sensors for global positioning while vehicles and drones are also equipped with IMU sensors. LiDAR point clouds are visualized in Figure 3 (b). Sensors from vehicles, drones, and RSUs are annotated with different colors. LiDAR point clouds of different types of agents complement each other, forming more complete point clouds of the scene.

Table 2: Sensor configurations for vehicles, RSUs, and drones.

| Agent | LiDAR | Cameras | Other Sensors |
|---|---|---|---|
| Vehicle | 1× 360° 64-channel LiDAR; 20 Hz rotation; Vertical FOV +10°/-38° | 6× surround-view cameras; FOV 110°; 1280× 720 resolution | GNSS, IMU |
| RSU | 1× 360° 64-channel LiDAR; 20 Hz rotation; Vertical FOV +10°/-38° | 4× cameras (front, left, right, back); FOV 110°; 1280× 720 resolution | GNSS |
| Drone | 1× 360° 64-channel LiDAR; 20 Hz rotation; Vertical FOV -30°/-90° | 1× downward-facing camera; FOV 110°; 1280× 720 resolution | GNSS, IMU |

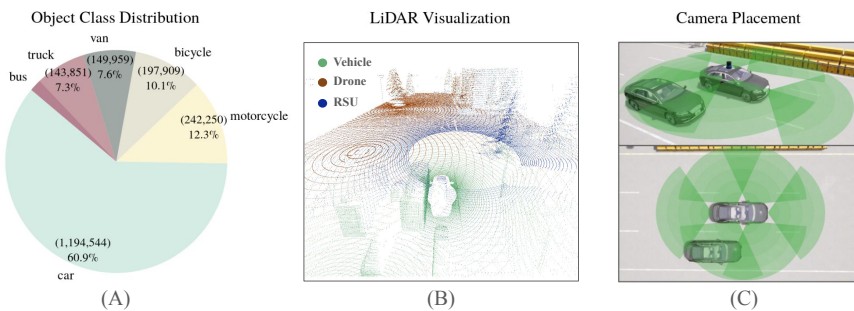

Figure 3: (a) The percentage of each scene in the dataset; (b) LiDAR point clouds visualization for each agent types, and (c) the camera/LiDAR placement for vehicular agents.

### 3.4 DATA ANNOTATION AND DOWNSTREAM TASKS

CARLA simulator provides accurate ground truth labels for all the objects in the scene, including the 3D bounding boxes, semantic segmentation, depth map, and semantic LiDAR point clouds. Using such labels, we provide 3D bounding boxes annotations for 3D object detection, segmentation map annotations for BEV semantic segmentation, depth map annotations for depth estimation, and tracking annotations for 3D multi-object tracking. The data and annotations are gathered in a synchronized manner across all agents in the scene, enabling the collaborative perception tasks. The dataset is split into training, validation, and test sets that contain 2.19, 1.02, and 3.52 hours of driving sequences, respectively. We annotate 6 categories of common objects in the driving scenarios including cars, motorcycles, bicycles, vans, trucks, and buses, forming a total of 1,961,484 annotated objects. The statistics of the dataset are summarized in Figure 3 (a).

## 4 BENCHMARK

In this section, we benchmark multi-agent collaborative perception algorithms on our proposed AirV2X-Perception dataset. We carefully select six representative algorithms that showcase different approaches to collaborative perception challenges, with particular attention to their applicability in complex, heterogeneous, multi-agent scenarios involving ground vehicles, infrastructure, and aerial perspectives. Our evaluation spans 3D object detection, BEV semantic segmentation, and computational efficiency across diverse environmental conditions and agent configurations, providing insights into both algorithmic performance and practical deployment considerations.

### 4.1 EXPERIMENTAL SETUP AND METHODOLOGY

The AirV2X-Perception dataset has three agent types: vehicles, roadside units (RSUs), and drones. To evaluate performance across this heterogeneous setting, we select representative collaborative perception algorithms: transformer-based V2XViT (Xu et al., 2022b); communication-efficient When2com (Liu et al., 2020) and Where2comm (Hu et al., 2022); BEV-optimized CoBEVT (Xu et al., 2022a); and heterogeneous-agent methods HEAL (Lu et al., 2024) and STAMP (Gao et al., 2025b). All experiments were conducted on RTX A6000 GPUs with PyTorch 2.6.0 and CUDA 12.6. To ensure fair comparison, we focus on LiDAR-based V2X perception—supported by most existing methods—and apply consistent training hyperparameters. Implementation details and reproducibility resources are available in the accompanying codebase[1].

### 4.2 PERFORMANCE OVERVIEW

We evaluate methods along three primary dimensions: 3D object detection accuracy, BEV semantic segmentation quality, and computational efficiency (peak GPU memory). For 3D detection, we use $[-140.8, 140.8] \times [-40, 40]$ meters around the ego vehicle and report mAP at 30% and 50% IoU (AP30, AP50). For BEV segmentation, we restrict the area to $[-64, 64] \times [-64, 64]$ meters and report mIoU. Following prior datasets such as Dair-V2X (Yu et al., 2022) and TUMTraf (Zimmer et al.,

---

[1]https://anonymous.4open.science/r/AirV2X-Perception-BBA7

2024), all vehicle types are treated as a single class for consistency. Peak GPU memory denotes the maximum usage during training per batch. Table 3 summarizes results across these dimensions.

The results show several trends. First, **heterogeneous agent collaboration methods (HEAL and STAMP) achieve superior performance** in object detection, with HEAL reaching the highest AP30 and AP50 scores of 49.2% and 45.5%. This indicates that explicitly modeling agent heterogeneity is advantageous in complex multi-agent scenarios such as those in our AirV2X-Perception dataset. Second, we observe a clear **trade-off between accuracy and efficiency**: while HEAL attains state-of-the-art detection results, it con-

Table 3: Performance overview of 3D object detection, BEV semantic segmentation, and peak GPU memory usage.

| Method | AP30↑ | AP50↑ | mIoU↑ | VRAM↓(GB) |
|---|---|---|---|---|
| When2com | 23.0 | 20.5 | 15.0 | **8.2** |
| CoBEVT | 42.9 | 29.8 | 33.9 | 38.1 |
| Where2comm | 44.8 | 37.0 | 29.2 | 10.2 |
| V2XViT | 46.4 | 39.1 | 32.7 | 43.5 |
| HEAL | **49.2** | **45.5** | **33.9** | 12.4 |
| STAMP | _47.9_ | _42.7_ | 27.7 | _10.1_ |

sumes 51% more memory than When2com, the most memory-efficient baseline. For semantic segmentation, both HEAL and CoBEVT achieve the best mIoU of 33.9%, with V2XViT close behind at 32.7%. The strong performance of STAMP and HEAL across both tasks underscores the importance of handling heterogeneous agent inputs in collaborative perception.

### 4.3 PERFORMANCE ANALYSIS ACROSS ENVIRONMENTAL CONDITIONS

Table 4: 3D object detection results in different scenarios, including lighting conditions (day, dusk, or night), environments (urban or rural), and weather conditions (rainy, foggy, cloudy, or clear).

| Method | Day AP30 | Day AP50 | Dusk AP30 | Dusk AP50 | Night AP30 | Night AP50 | Urban AP30 | Urban AP50 | Rural AP30 | Rural AP50 | Rainy AP30 | Rainy AP50 | Foggy AP30 | Foggy AP50 | Cloudy AP30 | Cloudy AP50 | Clear AP30 | Clear AP50 |
|---|---|---|---|---|---|---|---|---|---|---|---|---|---|---|---|---|---|---|
| When2com | 21.5 | 20.8 | 24.8 | 20.6 | 15.4 | 15.0 | 25.3 | 22.4 | 16.1 | 15.5 | 17.2 | 16.4 | 32.7 | 26.4 | 25.3 | 22.4 | 22.7 | 22.1 |
| CoBEVT | 37.1 | 24.2 | 48.4 | 35.1 | 10.5 | 2.0 | 51.3 | 37.4 | 13.8 | 3.5 | 15.7 | 4.7 | 27.7 | 13.6 | 51.3 | 37.4 | 49.4 | 35.7 |
| Where2comm | 42.5 | 35.7 | 47.9 | 38.6 | 23.4 | 21.4 | 48.6 | 40.3 | 32.6 | 26.0 | 35.0 | 27.5 | 31.4 | 26.0 | 48.6 | 40.3 | 45.4 | 39.3 |
| V2XViT | 46.8 | 41.1 | 47.8 | 39.0 | 19.3 | 16.2 | 52.1 | 43.8 | 28.7 | 25.1 | 32.3 | 28.6 | 38.4 | 31.3 | 52.1 | 43.8 | 50.7 | 44.3 |
| HEAL | **49.2** | **46.9** | **49.9** | **44.7** | **32.8** | **31.7** | 52.8 | **48.9** | **36.8** | **34.1** | **38.5** | **35.7** | **45.6** | **39.3** | 52.8 | **48.9** | **54.8** | **52.8** |
| STAMP | 48.6 | 41.6 | 48.0 | 39.7 | 19.8 | 16.9 | **52.8** | 44.5 | 28.6 | 25.7 | 32.0 | 29.2 | 38.5 | 31.6 | **52.8** | 44.5 | 52.1 | 45.0 |

To evaluate real-world applicability, we analyze algorithm performance across environmental factors: lighting (day, dusk, night), scene type (urban or rural), and weather (rainy, foggy, cloudy, clear). As shown in Table 4, **all methods are strongly affected by lighting**, with large degradation at night. CoBEVT suffers the steepest drop, with nighttime AP30 (10.5%). In contrast, HEAL shows the highest resilience, retaining 65.7% of its daytime AP30 at night (32.8% vs. 49.2%). **Scene context** also matters: urban scenes consistently outperform rural ones. This gap is largest for CoBEVT (51.3% vs. 13.8%, a 73.1% reduction) and smallest for HEAL (52.8% vs. 36.8%, a 30.3% reduction), suggesting heterogeneous fusion particularly benefits rural environments.

For **weather**, CoBEVT degrades notably in fog (27.7% vs. 49.4% in clear), while HEAL remains steadier across conditions. Rain impacts all models, though HEAL and STAMP remain more robust than earlier methods like When2com and CoBEVT. Overall, these results highlight that collaborative perception methods differ greatly in robustness to environmental variation, underscoring the need for models resilient across diverse conditions.

### 4.4 IMPACT OF DRONE NAVIGATION STRATEGIES

A unique aspect of our AirV2X-Perception dataset is the incorporation of aerial agents (drones) with distinct navigation strategies: hover, patrol, and escort. We evaluate how these navigation strategies affect perception performance in Table 5. We can observe that hover mode presents the greatest challenge for all methods, possibly due to the limited range covered by static drones. Patrol mode yields the best results across algorithms, particularly for CoBEVT (64.2% AP30). Escort mode shows intermediate performance, with HEAL (41.8%

Table 5: Object detection results with different navigation strategies for drones, including hover, patrol, and escort.

| Method | Hover AP30 | Hover AP50 | Patrol AP30 | Patrol AP50 | Escort AP30 | Escort AP50 |
|---|---|---|---|---|---|---|
| When2com | 15.4 | 15.0 | 21.7 | 21.0 | 25.0 | 20.6 |
| CoBEVT | 10.5 | 2.0 | **64.2** | **54.3** | 21.5 | 9.0 |
| Where2comm | 23.4 | 21.4 | 57.8 | 48.4 | 31.9 | 25.7 |
| V2XViT | 19.3 | 16.2 | 58.8 | 50.9 | 34.7 | 28.8 |
| HEAL | **32.8** | **31.7** | 56.1 | 53.8 | **41.8** | **36.6** |
| STAMP | 19.8 | 16.9 | 59.8 | 51.7 | 34.7 | 29.2 |

AP30) outperforming V2XViT and STAMP (both 34.7% AP30), suggesting that more sophisticated fusion mechanisms better handle adaptive trajectories. These findings emphasize that drone navigation strategies should be coordinated with perception algorithm capabilities to optimize collaborative perception systems.

### 4.5 SEMANTIC SEGMENTATION PERFORMANCE ANALYSIS

The experiemental results of BEV semantic segmentation performance across environmental conditions and drone navigation strategies are shown in Table 6 and Table 7. Segmentation performance is consistent across environmental conditions. V2XViT's segmentation performance varies by only 6.6% points between its best (cloudy at 36.4%) and worst (foggy at 28.8%) conditions, compared to a 32.8% point spread in detection between urban settings (52.1%) and night scenarios (19.3%). For various drone navigation strategies, patrol mode yields the highest performance across all methods, with V2XViT reach-

Table 6: Semantic segmentation performance (mIoU) across various drone navigation strategies.

| Method | Hover | Patrol | Escort |
|---|---|---|---|
| When2com | 15.1 | 14.8 | 15.8 |
| CoBEVT | 34.3 | 40.3 | 28.4 |
| Where2comm | 31.7 | 33.8 | 27.6 |
| V2XViT | 32.7 | **40.5** | **38.9** |
| HEAL | **34.3** | 40.3 | 38.4 |
| STAMP | 26.3 | 32.1 | 30.7 |

ing 40.5% and both CoBEVT and HEAL achieving 40.3%, consistent with detection results.

Table 7: Segmentation performance (mIoU) across various lighting, environments, and weathers.

| Method | Day | Dusk | Night | Urban | Rural | Rainy | Foggy | Cloudy | Clear |
|---|---|---|---|---|---|---|---|---|---|
| When2com | 16.2 | 14.3 | 15.1 | 17.9 | 13.6 | 11.5 | 14.0 | 17.9 | 16.4 |
| CoBEVT | 33.1 | 34.2 | 34.3 | 33.7 | 27.0 | 28.3 | **30.7** | 33.7 | 32.7 |
| Where2comm | 32.8 | **36.5** | 31.7 | 31.6 | **28.6** | 24.1 | 28.7 | 31.6 | 30.0 |
| V2XViT | **34.7** | 35.0 | **32.7** | **36.4** | 28.1 | **29.7** | 28.8 | **36.4** | **33.9** |
| HEAL | 33.1 | 34.2 | 34.3 | 33.7 | 27.0 | 28.3 | **30.7** | 33.7 | 32.7 |
| STAMP | 26.1 | 33.5 | 26.4 | 36.1 | 25.1 | 22.2 | 27.4 | 36.1 | 33.5 |

### 4.6 IMPACT OF DEGRADED LiDAR SENSOR CONFIGURATIONS

To evaluate robustness under resource-constrained scenarios with degraded LiDAR configurations. While our primary benchmark uses 64-channel LiDAR, we collect data using 16- and 32-channel configurations to simulate cost-effective deployments, particularly relevant for aerial agents. Results in Table 8 show performance degradation across all methods with degraded LiDAR. However, simpler methods (When2com, CoBEVT) show better robustness to sensor degradation, maintaining

Table 8: Object detection performance (AP30) with different LiDAR channel configurations.

| Method | 64-ch | 32-ch | 16-ch |
|---|---|---|---|
| When2com | 23.0 | 15.2 ↓7.8 | 5.6 ↓17.4 |
| CoBEVT | 42.9 | 32.8 ↓10.1 | 15.5 ↓27.4 |
| Where2comm | 44.8 | 31.5 ↓13.3 | 21.8 ↓23.0 |
| V2XViT | 46.4 | 29.9 ↓16.5 | 17.0 ↓29.4 |
| STAMP | 47.9 | 14.1 ↓33.8 | 11.1 ↓36.8 |
| HEAL | 49.2 | 14.9 ↓34.3 | 12.2 ↓37.0 |

24.3% and 36.1% of baseline performance, while heterogeneous methods (STAMP, HEAL) experience more severe drops to 23.2% and 24.8%. This suggests that sophisticated fusion strategies may require adaptation for low-quality sensors, highlighting a trade-off between accuracy and robustness.

### 4.7 IMPACT OF TEMPORAL AND SPATIAL ERRORS

Table 9 summarizes the robustness of different V2X perception methods under temporal asynchronization and localization/direction errors. Temporal asynchronization generally reduces performance, but training with asynchronous data mitigates the degradation. CoBEVT shows a slight improvement (+1.0%) under asynchronous training, suggesting that temporal misalignment may act as implicit data augmentation. Localization and direction errors further highlight system fragility. Performance consistently drops as noise increases, with localization errors (0.2m) causing more severe degradation (3.4–5.6%) than comparable direction errors (0.2°) at 2.2–3.6%. STAMP and HEAL exhibit higher sensitivity, indicating that sophisticated feature fusion mechanisms require

precise spatial alignment. Overall, these findings emphasize that real-world V2X systems must account for both temporal and spatial inconsistencies. Robust training strategies and accurate pose estimation are critical for maintaining performance under realistic deployment conditions.

Table 9: Object detection performance (AP30) of different V2X methods under temporal asynchronization and localization/direction error simulations. "Sync/Async" denotes synchronized training and asynchronous evaluation; "Async/Async" denotes asynchronous training and evaluation.

| Method | Baseline | Temporal Asynchronization | | Localization & Direction Errors | | | |
|--------|----------|-------------|--------------|-----------------|-----------------|---------------------|---------------------|
| | | Sync/Async | Async/Async | $\sigma = 0.1$m | $\sigma = 0.2$m | $\sigma = 0.2°$ | $\sigma = 0.4°$ |
| When2com | 23.0 | 20.3 ↓2.7 | 22.5 ↓0.5 | 20.3 ↓2.7 | 18.5 ↓4.5 | 19.8 ↓3.2 | 17.1 ↓5.9 |
| CoBEVT | 42.9 | 41.0 ↓1.9 | 43.9 ↑1.0 | 41.0 ↓1.9 | 39.2 ↓3.7 | 40.4 ↓2.5 | 37.8 ↓5.1 |
| Where2comm | 44.8 | 43.2 ↓1.6 | 43.2 ↓1.6 | 43.2 ↓1.6 | 41.4 ↓3.4 | 42.6 ↓2.2 | 40.2 ↓4.6 |
| V2XViT | 46.4 | 44.0 ↓2.4 | 45.1 ↓1.3 | 44.0 ↓2.4 | 41.7 ↓4.7 | 43.2 ↓3.2 | 40.1 ↓6.3 |
| STAMP | 49.2 | 46.9 ↓2.3 | 48.6 ↓0.6 | 46.9 ↓2.3 | 44.1 ↓5.1 | 46.1 ↓3.1 | 42.7 ↓6.5 |
| HEAL | 47.9 | 45.1 ↓2.8 | 45.2 ↓2.7 | 45.1 ↓2.8 | 42.3 ↓5.6 | 44.3 ↓3.6 | 40.8 ↓7.1 |

## 5 DISCUSSION

Our comprehensive evaluation of multi-agent collaborative perception algorithms reveals several key insights that can guide future research in both algorithmic improvement and dataset design.

### 5.1 ALGORITHMIC IMPROVEMENT

The AirV2X-Perception dataset presents a particularly demanding benchmark due to its **heterogeneous collaboration requirements** among vehicles, roadside units (RSUs), and drones, each with distinct sensing capabilities and perspectives. Our results demonstrate that methods specifically designed for heterogeneous agents (HEAL and STAMP) consistently outperform conventional approaches, particularly in challenging environmental conditions. This underscores the importance of algorithms that can effectively integrate information from diverse sensing modalities and viewpoints.

**Computational efficiency for large-scale collaborative perception** represents another significant challenge. Real-world deployments may involve dozens or hundreds of interconnected agents, yet several current approaches employ self-attention mechanisms that scale quadratically with agent count. The substantial variation in memory requirements across methods (8.2GB for When2com versus 43.5GB for V2XViT) highlights the need for algorithms that maintain perceptual accuracy while scaling efficiently to large agent networks.

Furthermore, **performance robustness across diverse environmental conditions** remains problematic. Our analysis shows substantial performance degradation in challenging scenarios, with even the best-performing method (HEAL) experiencing a 34% reduction in accuracy during nighttime operations. Developing environment-invariant collaborative perception systems that maintain consistent performance across all conditions constitutes a critical research direction.

### 5.2 FUTURE DATASET DESIGN

While the AirV2X-Perception dataset offers a comprehensive benchmark across diverse environments and agent configurations, **real-world deployment introduces additional complexities not fully captured in simulation**. Future research should emphasize datasets that incorporate realistic sensor noise, communication constraints, and environmental variations, enabling more reliable evaluation under authentic conditions and improving transferability from simulation to practice.

In real applications, perception informs decision-making, which in turn influences subsequent perceptions. **Closed-loop evaluation frameworks** that capture this feedback cycle would yield deeper insights into long-term performance beyond single-frame accuracy. Finally, future datasets should **include safety-critical edge cases** such as accidents, road blockages, construction zones, and extreme weather, as these scenarios are especially important for evaluating system robustness and ensuring safe autonomous driving.

**Ethics Statement.** This work relies entirely on simulated environments (CARLA and AirSim), ensuring no personal or sensitive real-world data is involved. All released data are synthetic sensor outputs from virtual vehicles, drones, and RSUs. The authors affirm compliance with the ICLR Code of Ethics and uphold the principles of scientific integrity, transparency, and responsible stewardship.

**Reproducibility Statement.** We release the full AirV2X-Perception dataset, benchmark splits, and accompanying codebase with training and evaluation scripts available at https://anonymous.4open.science/r/AirV2X-Perception-BBA7. All experiments specify hardware/software details, hyperparameters, and fixed random seeds, and we benchmarked established baselines using their official or verified implementations. Together, these resources enable independent researchers to reproduce our results and fairly compare future methods.

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

APPENDIX

## A    EXPERIMENTS DETAILS

All experiments were conducted under consistent hardware and software conditions using RTX A6000 GPUs, PyTorch 2.6.0, CUDA 12.6, and SpConv 2.1. Models were trained with a batch size of 4 using the Adam optimizer with an initial learning rate of 0.002 for 20 epochs. We trained and evaluated all models following their official implementations (or unofficial implementations where official ones were unavailable). Random seeds were fixed across all experiments to ensure reproducibility. For additional implementation details, please refer to our open-source codebase[2].

## B    ABLATION STUDIES FOR DIFFERENT NUMBERS AND TYPES OF AGENTS.

Table A1: Object Detection Results by Lighting Conditions Across Different Agent Combinations.

| Method | Overall | | Daytime | | Dusk | | Nighttime | |
|---|---|---|---|---|---|---|---|---|
| | AP30 | AP50 | AP30 | AP50 | AP30 | AP50 | AP30 | AP50 |
| **Vehicle + Infra + Drone** | | | | | | | | |
| When2comm | 23.0 | 20.5 | 21.5 | 20.8 | 24.8 | 20.6 | 15.4 | 15.0 |
| CoBEVT | 42.9 | 29.8 | 37.1 | 24.2 | 48.4 | 35.1 | 10.5 | 2.0 |
| Where2comm | 44.8 | 37.0 | 42.5 | 35.7 | 47.9 | 38.6 | 23.4 | 21.4 |
| V2XViT | 46.4 | 39.1 | 46.8 | 41.1 | 47.8 | 39.0 | 19.3 | 16.2 |
| HEAL | 49.2 | 45.5 | 49.2 | 46.9 | 49.9 | 44.7 | 32.8 | 31.7 |
| STAMP | 47.9 | 42.7 | 47.6 | 41.6 | 48.0 | 39.7 | 19.8 | 16.9 |
| **Vehicle + Infra** | | | | | | | | |
| When2comm | 16.2 6.8↓ | 16.1 4.4↓ | 16.4 5.1↓ | 12.6 8.2↓ | 19.0 5.8↓ | 12.1 8.5↓ | 7.9 7.6↓ | 7.2 7.9↓ |
| CoBEVT | 35.1 7.8↓ | 21.7 8.1↓ | 28.4 8.7↓ | 19.4 4.8↓ | 42.9 5.5↓ | 31.2 4.0↓ | 9.2 1.3↓ | 0.7 1.3↓ |
| Where2comm | 37.4 7.4↓ | 32.9 4.0↓ | 34.5 8.0↓ | 31.6 4.2↓ | 41.3 6.6↓ | 35.0 3.6↓ | 17.6 5.8↓ | 14.8 6.5↓ |
| V2XViT | 40.5 5.9↓ | 32.8 6.3↓ | 43.6 3.2↓ | 35.1 6.0↓ | 43.3 4.5↓ | 35.4 3.5↓ | 12.7 6.6↓ | 12.3 4.0↓ |
| HEAL | 45.3 3.9↓ | 42.6 3.0↓ | 44.5 4.7↓ | 43.5 3.4↓ | 46.4 3.5↓ | 39.5 5.2↓ | 22.8 10.0↓ | 16.7 15.0↓ |
| STAMP | 45.0 2.9↓ | 40.8 1.9↓ | 45.5 2.0↓ | 40.0 1.6↓ | 45.9 2.1↓ | 38.0 1.7↓ | 18.8 1.0↓ | 15.0 1.9↓ |
| **Vehicle only** | | | | | | | | |
| When2comm | 11.3 11.7↓ | 1.2 19.3↓ | 10.6 10.9↓ | 0.8 20.0↓ | 12.0 12.8↓ | 1.8 18.9↓ | 3.0 12.4↓ | 0.4 14.7↓ |
| CoBEVT | 23.7 19.2↓ | 8.0 21.8↓ | 25.4 11.7↓ | 5.1 19.1↓ | 25.7 22.7↓ | 5.5 29.6↓ | 5.3 5.2↓ | 0.5 1.5↓ |
| Where2comm | 24.5 20.3↓ | 5.3 31.7↓ | 23.1 19.4↓ | 4.7 31.0↓ | 26.1 21.8↓ | 5.8 32.7↓ | 7.0 16.4↓ | 2.4 18.9↓ |
| V2XViT | 28.5 17.9↓ | 9.3 29.8↓ | 26.8 20.0↓ | 9.6 31.4↓ | 29.5 18.2↓ | 9.2 29.8↓ | 11.2 8.0↓ | 1.8 14.4↓ |
| HEAL | 32.4 16.9↓ | 12.9 32.7↓ | 29.7 19.5↓ | 13.4 33.5↓ | 33.3 16.6↓ | 14.0 30.7↓ | 12.5 20.3↓ | 1.7 30.0↓ |
| STAMP | 31.0 16.9↓ | 13.4 29.3↓ | 28.6 18.9↓ | 13.8 27.8↓ | 33.1 14.9↓ | 12.7 27.0↓ | 10.2 9.6↓ | 1.8 15.1↓ |

This section analyzes the impact of different agent types (vehicles, infrastructure, and drones) on perception performance across various environmental conditions and scenarios.

As shown in Table A1, when examining overall performance, removing drone agents leads to a moderate performance drop, while using only vehicle agents results in a severe degradation. The performance decline is particularly pronounced in AP50 metrics, where vehicle-only configurations suffer decreases of up to 32.7% for models like HEAL. This suggests that infrastructure agents provide critical spatial information that complements vehicle perspectives. For nighttime scenarios, the contribution of drone agents becomes especially valuable, with their removal causing HEAL's AP50 score to drop by 15.0%. This highlights drones' ability to maintain visibility in low-light conditions where ground-based agents struggle. The aerial perspective provided by drones offers a strategic advantage in maintaining perception reliability across varying environmental conditions, particularly in scenarios with compromised lighting.

Table A2: Object Detection Results by Environments Across Different Agent Combinations.

| Method | Urban | | Rural | |
|---|---|---|---|---|
| | AP30 | AP50 | AP30 | AP50 |
| **Vehicle + Infra + Drone** | | | | |
| When2comm | 25.3 | 22.4 | 16.1 | 15.5 |
| CoBEVT | 51.3 | 37.4 | 13.8 | 3.5 |
| Where2comm | 48.6 | 40.3 | 32.6 | 26.0 |
| V2XViT | 52.1 | 43.8 | 28.7 | 25.1 |
| HEAL | 52.8 | 48.9 | 36.8 | 34.1 |
| STAMP | 52.8 | 44.5 | 28.6 | 25.7 |
| **Vehicle + Infra** | | | | |
| When2comm | 20.8 4.4↓ | 15.0 7.4↓ | 12.8 3.3↓ | 10.0 5.5↓ |
| CoBEVT | 47.2 4.1↓ | 29.4 8.1↓ | 8.9 4.8↓ | 3.3 0.1↓ |
| Where2comm | 40.1 8.5↓ | 36.4 3.9↓ | 26.5 6.1↓ | 23.0 3.0↓ |
| V2XViT | 44.6 7.6↓ | 40.8 3.1↓ | 22.9 5.9↓ | 16.5 8.6↓ |
| HEAL | 52.1 0.8↓ | 46.5 2.5↓ | 33.3 3.4↓ | 30.5 3.6↓ |
| STAMP | 52.3 0.6↓ | 43.4 1.0↓ | 25.9 2.7↓ | 23.0 2.7↓ |
| **Vehicle only** | | | | |
| When2comm | 13.3 12.0↓ | 1.4 21.0↓ | 4.5 11.6↓ | 0.4 15.1↓ |
| CoBEVT | 32.0 19.3↓ | 6.3 31.1↓ | 6.7 7.1↓ | 3.6 0.2↑ |
| Where2comm | 27.5 21.1↓ | 6.1 34.2↓ | 14.0 18.6↓ | 2.7 23.4↓ |
| V2XViT | 33.9 18.3↓ | 10.8 33.0↓ | 10.5 18.3↓ | 4.7 20.4↓ |
| HEAL | 37.8 15.0↓ | 16.5 32.4↓ | 12.5 24.2↓ | 4.6 29.5↓ |
| STAMP | 36.4 16.5↓ | 16.2 28.3↓ | 11.9 16.8↓ | 3.3 22.4↓ |

---

[2]https://github.com/taco-group/AirV2X-Perception

Table A2 shows that in urban settings, HEAL and STAMP demonstrate remarkable robustness to drone removal, with minimal AP30 drops of 0.8 and 0.6% respectively, suggesting that the dense infrastructure in urban areas can partially compensate for the elevated perspective that drones provide. The multiple perception points available from infrastructure agents in urban environments appear sufficient to maintain reliable detection performance even without aerial data streams. Rural environments tell a different story, with every agent type providing crucial information. The absence of dense infrastructure in rural settings makes drone perspectives particularly valuable, as evidenced by performance drops of 2.7%-6.1% AP30 when drones are removed. The vehicle-only configuration performs drastically worse in rural settings, with HEAL experiencing a 29.5% decrease in AP50, highlight-

Table A3: Object Detection Results by Drones' Navigation Strategies Across Different Agent Combinations.

| Method | Hover | | Patrol | | Escort | |
|---|---|---|---|---|---|---|
| | AP30 | AP50 | AP30 | AP50 | AP30 | AP50 |
| **Vehicle + Infra + Drone** | | | | | | |
| When2comm | 15.4 | 15.0 | 21.7 | 21.0 | 25.0 | 20.6 |
| CoBEVT | 10.5 | 2.0 | 64.2 | 54.3 | 21.5 | 9.0 |
| Where2comm | 23.4 | 21.4 | 57.8 | 48.4 | 31.9 | 25.7 |
| V2XViT | 19.3 | 16.2 | 58.8 | 50.9 | 34.7 | 28.8 |
| HEAL | 32.8 | 31.7 | 56.1 | 53.8 | 41.8 | 36.6 |
| STAMP | 19.8 | 16.9 | 59.8 | 51.7 | 34.7 | 29.2 |
| **Vehicle + Infra** | | | | | | |
| When2comm | 7.9 7.6↓ | 7.2 7.9↓ | 16.4 5.3↓ | 13.8 7.2↓ | 20.0 5.0↓ | 15.6 5.0↓ |
| CoBEVT | 8.2 2.3↓ | 0.7 1.3↓ | 57.1 7.1↓ | 45.6 8.7↓ | 15.1 6.4↓ | 11.3 2.3↑ |
| Where2comm | 17.6 5.8↓ | 14.8 6.5↓ | 52.4 5.4↓ | 42.3 6.0↓ | 28.8 3.0↓ | 21.3 4.4↓ |
| V2XViT | 12.7 6.6↓ | 12.3 4.0↓ | 55.8 3.0↓ | 44.9 6.1↓ | 28.9 5.8↓ | 20.4 8.4↓ |
| HEAL | 22.8 10.0↓ | 16.7 15.0↓ | 57.6 1.5↑ | 49.0 4.8↓ | 41.3 0.5↓ | 36.4 0.2↓ |
| STAMP | 18.8 1.0↓ | 15.0 1.9↓ | 58.6 1.2↓ | 51.1 0.6↓ | 35.1 0.5↑ | 27.2 2.0↓ |
| **Vehicle only** | | | | | | |
| When2comm | 3.0 12.4↓ | 0.4 14.7↓ | 16.1 5.6↓ | 1.9 19.2↓ | 6.5 18.5↓ | 0.8 19.8↓ |
| CoBEVT | 5.3 5.2↓ | 0.5 1.5↓ | 35.9 28.2↓ | 13.5 40.8↓ | 12.9 8.7↓ | 3.5 5.5↓ |
| Where2comm | 7.0 16.4↓ | 2.4 18.9↓ | 36.4 21.4↓ | 9.8 38.6↓ | 13.1 18.8↓ | 1.6 24.1↓ |
| V2XViT | 11.2 8.0↓ | 1.8 14.4↓ | 41.1 17.8↓ | 15.3 35.6↓ | 15.8 18.9↓ | 4.7 24.1↓ |
| HEAL | 12.5 20.3↓ | 1.7 30.0↓ | 45.5 10.6↓ | 24.9 28.9↓ | 18.0 23.8↓ | 5.8 30.8↓ |
| STAMP | 10.2 9.6↓ | 1.8 15.1↓ | 44.4 15.4↓ | 22.7 29.0↓ | 17.0 17.7↓ | 4.6 24.6↓ |

ing the challenges vehicles face in rural perception without additional perspectives. This significant disparity underscores how the collaborative perception benefits vary substantially based on environmental context.

Table A4: Object Detection Results by Weather Conditions Across Different Agent Combinations.

| Method | Rainy | | Foggy | | Cloudy | | Clear | |
|---|---|---|---|---|---|---|---|---|
| | AP30 | AP50 | AP30 | AP50 | AP30 | AP50 | AP30 | AP50 |
| **Vehicle + Infra + Drone** | | | | | | | | |
| When2comm | 17.2 | 16.4 | 32.7 | 26.4 | 25.3 | 22.4 | 22.7 | 22.1 |
| CoBEVT | 15.7 | 4.7 | 27.7 | 13.6 | 51.3 | 37.4 | 49.4 | 35.7 |
| Where2comm | 35.0 | 27.5 | 31.4 | 26.0 | 48.6 | 40.3 | 45.4 | 39.3 |
| V2XViT | 32.3 | 28.6 | 38.4 | 31.3 | 52.1 | 43.8 | 50.7 | 44.3 |
| HEAL | 38.5 | 35.7 | 45.6 | 39.3 | 52.8 | 48.9 | 54.8 | 52.8 |
| STAMP | 32.0 | 29.2 | 38.5 | 31.6 | 52.8 | 44.5 | 52.1 | 45.0 |
| **Vehicle + Infra** | | | | | | | | |
| When2comm | 11.0 6.2↓ | 9.5 6.8↓ | 28.9 3.8↓ | 20.0 6.4↓ | 20.8 4.4↓ | 15.0 7.4↓ | 18.5 4.3↓ | 16.9 5.1↓ |
| CoBEVT | 13.5 2.2↓ | 3.3 1.4↓ | 23.2 4.5↓ | 7.2 6.4↓ | 47.2 4.1↓ | 29.4 8.1↓ | 42.6 6.8↓ | 31.2 4.5↓ |
| Where2comm | 27.6 7.4↓ | 24.0 3.5↓ | 22.8 8.6↓ | 19.8 6.2↓ | 40.1 8.5↓ | 36.4 3.9↓ | 39.7 5.8↓ | 36.0 3.4↓ |
| V2XViT | 26.6 5.7↓ | 23.0 5.6↓ | 35.2 3.2↓ | 26.0 5.4↓ | 44.6 7.6↓ | 40.8 3.1↓ | 46.4 4.3↓ | 36.1 8.2↓ |
| HEAL | 36.4 2.1↓ | 33.8 1.9↓ | 43.8 1.8↓ | 38.6 0.6↓ | 50.1 2.8↓ | 46.5 2.5↓ | 50.6 4.3↓ | 46.6 6.2↓ |
| STAMP | 30.2 1.8↓ | 26.9 2.4↓ | 36.7 1.8↓ | 31.1 0.5↓ | 52.3 0.6↓ | 43.4 1.0↓ | 50.5 1.5↓ | 44.1 0.9↓ |
| **Vehicle only** | | | | | | | | |
| When2comm | 5.1 12.1↓ | 0.4 16.0↓ | 8.3 24.4↓ | 1.2 25.2↓ | 13.3 12.0↓ | 1.4 21.0↓ | 14.3 8.4↓ | 1.1 21.0↓ |
| CoBEVT | 5.5 10.1↓ | 2.7 2.0↓ | 15.2 12.5↓ | 2.8 10.8↓ | 32.0 19.3↓ | 6.3 31.1↓ | 32.0 17.4↓ | 10.2 25.5↓ |
| Where2comm | 15.9 19.1↓ | 2.8 24.7↓ | 13.0 18.4↓ | 1.5 24.5↓ | 27.5 21.1↓ | 6.1 34.2↓ | 25.7 19.7↓ | 5.8 33.6↓ |
| V2XViT | 10.8 21.5↓ | 5.8 22.8↓ | 21.1 17.3↓ | 5.6 25.8↓ | 33.9 18.3↓ | 10.8 33.0↓ | 36.6 14.2↓ | 12.0 32.3↓ |
| HEAL | 12.2 26.3↓ | 5.0 30.7↓ | 21.0 24.6↓ | 8.0 31.3↓ | 37.8 15.0↓ | 16.5 32.4↓ | 38.3 16.6↓ | 19.1 33.6↓ |
| STAMP | 12.8 19.3↓ | 3.8 25.4↓ | 21.1 17.4↓ | 5.8 25.7↓ | 36.4 16.5↓ | 16.2 28.3↓ | 37.4 14.7↓ | 19.0 26.0↓ |

Table A3 displays the impact of different drone flight patterns on performance. In hover scenarios, drones provide essential overhead perspectives, with their removal causing significant decreases in AP50 (1.9%-15.0%). Vehicle-only configurations suffer catastrophic degradation in hover scenarios, with AP50 decreases of up to 30.0% for HEAL. For patrol scenarios, where drones follow predetermined routes, HEAL surprisingly shows a 1.5-point AP30 improvement when removing drones,

indicating potential conflicts between drone and ground agent information during this pattern. However, this anomaly is not reflected in AP50 metrics, where all models show performance drops. The escort pattern reveals interesting dynamics, with STAMP showing a slight 0.5% improvement in AP30 when removing drones, suggesting that closely following drone patterns may sometimes introduce redundant or conflicting information.

Table A5: Semantic Segmentation Results by Lighting Conditions, Environments, and Weather Conditions Across Different Agent Combinations.

| Method | Overall | Day | Dusk | Night | Urban | Rural | Rainy | Foggy | Cloudy | Clear |
|---|---|---|---|---|---|---|---|---|---|---|
| **Vehicle + Infra + Drone** | | | | | | | | | | |
| When2comm | 15.0 | 16.2 | 14.3 | 15.1 | 17.9 | 13.6 | 11.5 | 14.0 | 17.9 | 16.4 |
| CoBEVT | 33.9 | 33.1 | 34.2 | 34.3 | 33.7 | 27.0 | 28.3 | 30.7 | 33.7 | 32.7 |
| Where2comm | 29.2 | 32.8 | 36.5 | 31.7 | 31.6 | 28.6 | 24.1 | 28.7 | 31.6 | 30.0 |
| V2XViT | 32.7 | 34.7 | 35.0 | 32.7 | 36.4 | 28.1 | 29.7 | 28.8 | 36.4 | 33.9 |
| HEAL | 33.9 | 33.1 | 34.2 | 34.3 | 33.7 | 27.0 | 28.3 | 30.7 | 33.7 | 32.7 |
| STAMP | 27.7 | 26.1 | 33.5 | 26.3 | 36.1 | 25.1 | 22.2 | 27.4 | 36.1 | 33.5 |
| **Vehicle + Infra** | | | | | | | | | | |
| When2comm | 13.6 1.3↓ | 14.7 1.4↓ | 13.1 1.3↓ | 13.8 1.3↓ | 16.3 1.6↓ | 12.3 1.3↓ | 10.4 1.0↓ | 12.7 1.3↓ | 16.3 1.6↓ | 14.9 1.5↓ |
| CoBEVT | 27.9 6.1↓ | 27.1 6.0↓ | 28.0 6.1↓ | 28.2 6.2↓ | 27.6 6.1↓ | 22.1 4.8↓ | 23.3 5.1↓ | 25.2 5.6↓ | 27.6 6.1↓ | 26.8 5.9↓ |
| Where2comm | 24.0 5.2↓ | 26.9 5.9↓ | 29.9 6.6↓ | 26.0 5.7↓ | 25.9 5.7↓ | 23.5 5.2↓ | 19.8 4.3↓ | 23.6 5.2↓ | 25.9 5.7↓ | 24.6 5.4↓ |
| V2XViT | 26.8 5.9↓ | 28.5 6.3↓ | 28.7 6.3↓ | 26.8 5.9↓ | 29.8 6.5↓ | 23.1 5.0↓ | 24.4 5.3↓ | 23.6 5.2↓ | 29.8 6.5↓ | 27.8 6.1↓ |
| HEAL | 27.9 6.1↓ | 27.1 6.0↓ | 28.1 6.1↓ | 28.1 6.2↓ | 27.7 6.1↓ | 22.1 4.8↓ | 23.3 5.1↓ | 25.2 5.5↓ | 27.7 6.1↓ | 26.8 5.9↓ |
| STAMP | 22.7 5.0↓ | 21.4 4.7↓ | 27.5 6.0↓ | 21.5 4.7↓ | 29.6 6.5↓ | 20.6 4.5↓ | 18.3 4.0↓ | 22.5 4.9↓ | 29.6 6.5↓ | 27.5 6.0↓ |
| **Vehicle only** | | | | | | | | | | |
| When2comm | 12.5 2.4↓ | 13.6 2.6↓ | 12.1 2.3↓ | 12.7 2.4↓ | 15.0 2.9↓ | 11.4 2.2↓ | 9.6 1.8↓ | 11.8 2.2↓ | 15.0 2.9↓ | 13.8 2.6↓ |
| CoBEVT | 23.7 10.2↓ | 23.2 9.9↓ | 23.9 10.3↓ | 24.0 10.3↓ | 23.6 10.1↓ | 18.9 8.1↓ | 19.8 8.5↓ | 21.5 9.2↓ | 23.6 10.1↓ | 22.9 9.8↓ |
| Where2comm | 20.5 8.8↓ | 22.9 9.8↓ | 25.6 10.9↓ | 22.2 9.5↓ | 22.1 9.5↓ | 20.1 8.6↓ | 16.9 7.3↓ | 20.1 8.6↓ | 22.1 9.5↓ | 21.0 9.0↓ |
| V2XViT | 22.9 9.8↓ | 24.3 10.4↓ | 24.4 10.5↓ | 22.9 9.8↓ | 25.4 10.9↓ | 19.7 8.4↓ | 20.7 8.9↓ | 20.2 8.6↓ | 25.4 10.9↓ | 23.7 10.2↓ |
| HEAL | 23.8 10.2↓ | 23.2 9.9↓ | 23.9 10.2↓ | 24.1 10.3↓ | 23.6 10.1↓ | 18.8 8.1↓ | 19.8 8.5↓ | 21.5 9.2↓ | 23.6 10.1↓ | 22.9 9.8↓ |
| STAMP | 19.4 8.3↓ | 18.3 7.9↓ | 23.5 10.0↓ | 18.4 7.9↓ | 25.3 10.9↓ | 17.5 7.5↓ | 15.6 6.7↓ | 19.2 8.2↓ | 25.3 10.9↓ | 23.4 10.1↓ |

Table A5 presents semantic segmentation performance across various conditions. For semantic segmentation, removing drone agents causes approximately 4.5%-6.5% decreases across most models and conditions. When comparing vehicle-only to vehicle+infrastructure, we observe an additional 3.0$-4.5$ drop, indicating that while drones provide valuable elevated perspectives for segmentation tasks, infrastructure agents also contribute significantly to boundary delineation and contextual understanding.

Table A6 reveals that drone flight patterns significantly impact semantic segmentation performance. The patrol pattern, where drones follow predetermined routes, yields the highest baseline performance across all models. Removing drones during patrol scenarios causes substantial performance drops (5.8%-7.3%). The vehicle-only configuration experiences the most severe degradation (9.6%-12.1%) in patrol scenarios, suggesting that this pattern provides complementary information that cannot be recovered from ground perspectives. For escort patterns, where drones follow specific vehicles, the performance drops are more consistent across models, with 5.0%-7.0% decreases when removing drones. The hover pattern shows

Table A6: Semantic Segmentation Results by Drones' Navigation Strategies Across Different Agent Combinations.

| Method | Hover | Patrol | Escort |
|---|---|---|---|
| **Vehicle + Infra + Drone** | | | |
| When2comm | 15.1 | 14.8 | 15.8 |
| CoBEVT | 34.3 | 40.3 | 28.4 |
| Where2comm | 31.7 | 33.8 | 27.6 |
| V2XViT | 32.7 | 40.5 | 38.9 |
| HEAL | 34.3 | 40.3 | 38.4 |
| STAMP | 26.3 | 32.1 | 30.7 |
| **Vehicle + Infra** | | | |
| When2comm | 13.8 1.3↓ | 13.4 1.4↓ | 14.4 1.4↓ |
| CoBEVT | 28.2 6.2↓ | 33.0 7.3↓ | 23.3 5.1↓ |
| Where2comm | 26.0 5.7↓ | 27.8 6.1↓ | 22.6 5.0↓ |
| V2XViT | 26.8 5.9↓ | 33.2 7.3↓ | 31.9 7.0↓ |
| HEAL | 28.1 6.2↓ | 33.1 7.3↓ | 31.5 6.9↓ |
| STAMP | 21.5 4.7↓ | 26.3 5.8↓ | 25.1 5.6↓ |
| **Vehicle only** | | | |
| When2comm | 12.7 2.4↓ | 12.5 2.4↓ | 13.2 2.5↓ |
| CoBEVT | 24.0 10.3↓ | 28.2 12.1↓ | 19.9 8.5↓ |
| Where2comm | 22.2 9.5↓ | 23.7 10.1↓ | 19.4 8.3↓ |
| V2XViT | 22.9 9.8↓ | 28.3 12.1↓ | 27.2 11.7↓ |
| HEAL | 24.1 10.3↓ | 28.2 12.1↓ | 26.9 11.5↓ |
| STAMP | 18.4 7.9↓ | 22.5 9.6↓ | 21.4 9.2↓ |

slightly less sensitivity to drone removal (4.7%-6.2%), suggesting that the aerial perspectives are limited in the hover mode because the drones are stationary.

In conclusion, our comprehensive ablation studies demonstrate that each agent type contributes uniquely to perception performance, with their relative importance varying across environmental conditions, scenarios, and perception tasks. Overall, both drone agents and infrastructure agents provide valuable perspectives multi-agent collaborative perception.

## C  DATASET VISUALIZATION

Figures A1 and A2 provide qualitative insight into the breadth and fidelity of the *AirV2X-Perception* dataset. Each scene is captured synchronously by heterogeneous sensing platforms—including road-side units (RSUs), connected vehicles, and drones. *(i)* The top panels in both figures highlight the raw RGB imagery acquired by the surround camera matrix on RSUs and vehicles. *(ii)* The middle panels display bird's-eye-view (BEV) camera image from the drone (left), a semantic BEV map (middle), and a perspective LiDAR point cloud with 3-D bounding boxes (right). *(iii)* The bottom panel displays the full LiDAR sweep onto the BEV plane axis-aligned bounding boxes. For the complete dataset, please refer to our open-sourced dataset link[3].

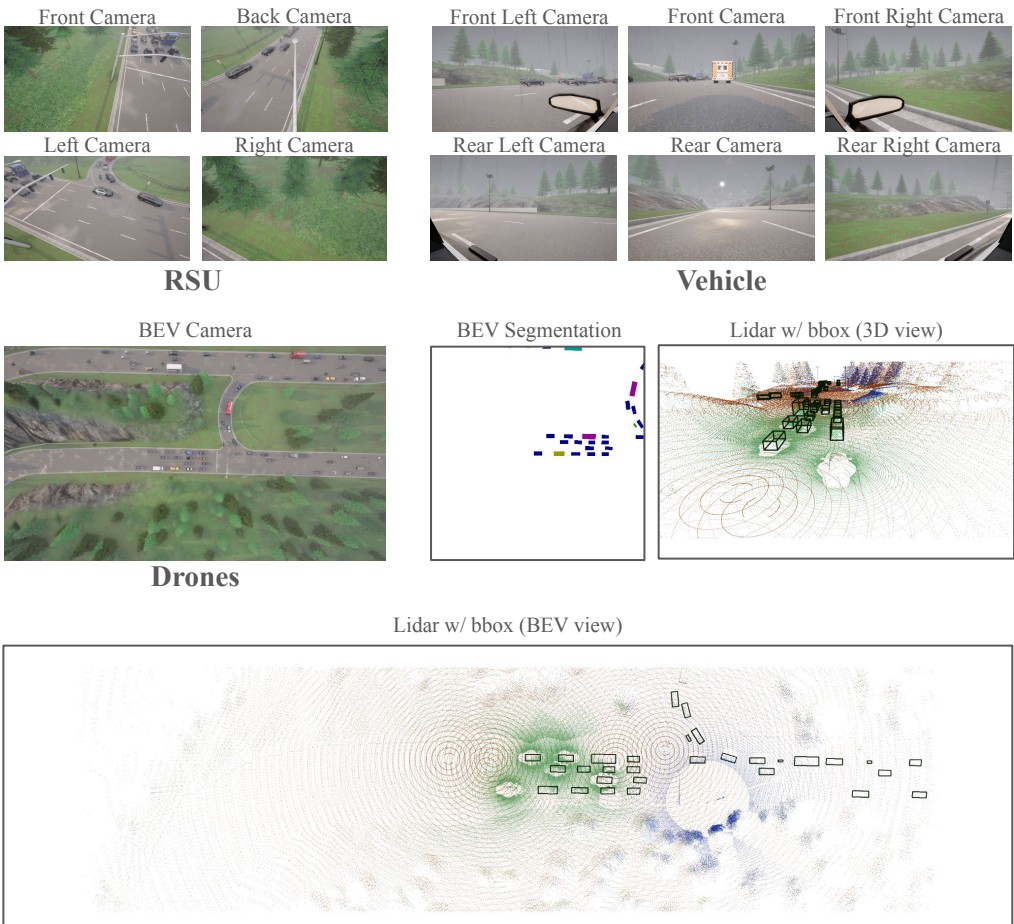

Figure A1: Visualization of some representative data of a single timestamp of the AirV2X-Perception dataset. Note that for each agent type from RSU, vehicle, and drone, only one agent is chosen for visualization.

---

[3] https://huggingface.co/datasets/xiangbog/AirV2X-Perception

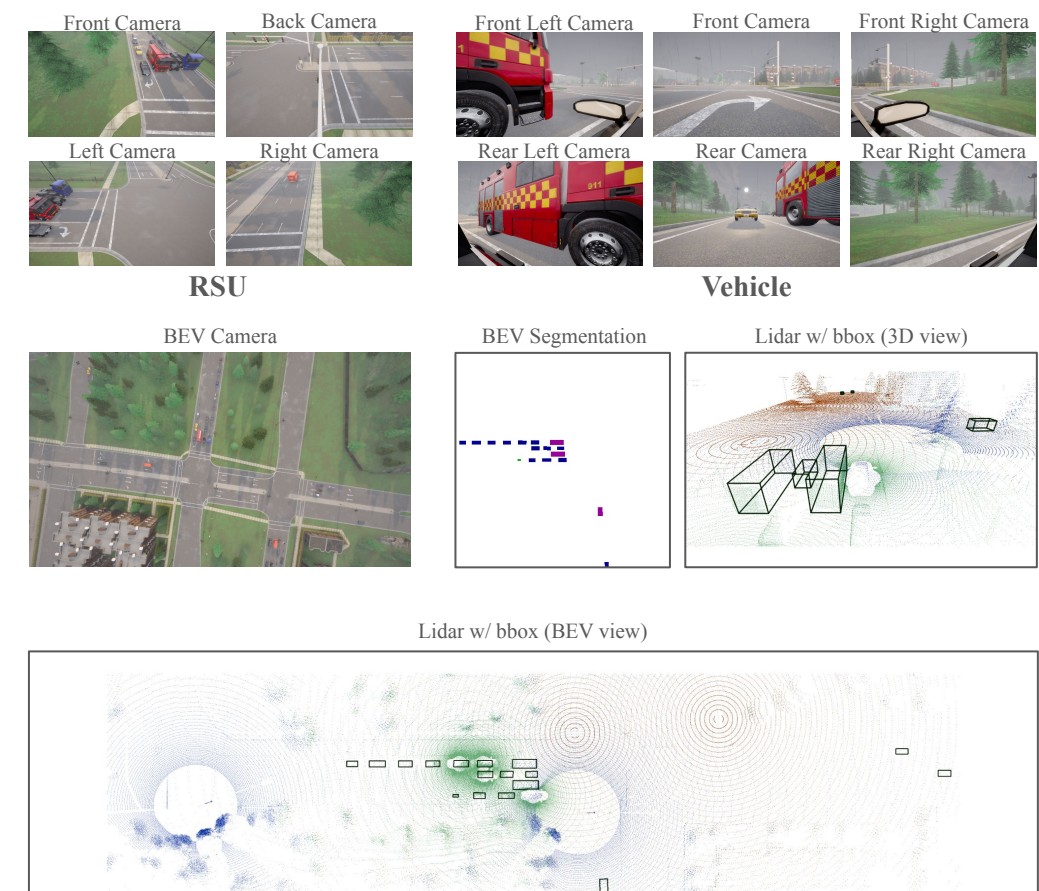

Figure A2: Visualization of some representative data of a single timestamp of the AirV2X-Perception dataset. Note that for each agent type from RSU, vehicle, and drone, only one agent is chosen for visualization.

# D BEYOND THE DATASET

## D.1 DRONE-ASSISTED VEHICLE-TO-EVERYTHING (V2X) FOR AUTONOMOUS DRIVING

The convergence of autonomous driving technologies with unmanned aerial vehicles (UAVs) presents compelling opportunities for intelligent transportation systems. The global drone industry is experiencing remarkable growth, with annual UAV shipments projected to reach 9.5 million units by 2029 (Hayes, 2024), corresponding to a market value of \$35-54 billion in 2024. Within the mobility domain, UAV-assisted logistics and transportation services alone are expected to double from \$5.3 billion in 2019 to \$11 billion by 2026 (Afrin et al., 2024). This proliferation of UAV technology is creating new possibilities for drone-assisted Vehicle-to-Everything (V2X) frameworks to enhance autonomous driving capabilities globally.

V2X communication, encompassing interactions between vehicles and their environment (other vehicles, infrastructure, pedestrians, networks), forms the foundation of connected autonomous driving. However, accommodating the growing number of connected devices and data-intensive services in vehicular networks challenges existing infrastructure (Wang & Zhang, 2025). Drone-assisted V2X offers a promising solution by integrating aerial drones as dynamic sensor platforms, communication relays, and edge computing nodes. UAVs effectively add a third dimension to V2X networks, enabling more comprehensive coverage and adaptive architectures than static ground infrastructure alone can provide.

## D.2 CURRENT RESEARCH TRENDS AND FUTURE PROJECTIONS

Research on drone-assisted vehicular networks has accelerated, exploring innovative protocol designs, efficient resource management, and energy-optimized operations for UAV nodes in V2X ecosystems. Current research prototypes demonstrate UAVs functioning as aerial base stations, relays, or cooperative sensing platforms in connected vehicle environments. Field trials show that drones can wirelessly connect isolated vehicle clusters, extend coverage in rural areas, and provide aerial perspectives to detect hazards beyond a vehicle's line-of-sight. Cooperative perception is emerging as a particularly valuable application—drones equipped with cameras or LiDAR can stream data to nearby vehicles, effectively enabling them to "see" around corners or beyond obstructions.

Looking ahead, sixth-generation (6G) wireless architectures are expected to natively support airborne communication nodes, facilitating real-time coordination between dense drone swarms and ground vehicles (Kavas-Torris et al., 2022). Researchers anticipate advanced UAV traffic management systems and dynamically reconfigurable airborne base stations that adapt to changing traffic conditions. Advancements in AI are projected to enhance multi-UAV collaboration, enabling autonomous drone swarms to optimize their positioning for network coverage and data collection. The trajectory of research suggests that drone-assisted V2X will evolve from today's experimental implementations to become a fundamental component of smart transportation within the decade.

## D.3 ADVANTAGES OVER TRADITIONAL RSUS AND VEHICLE-ONLY SYSTEMS

Drone-assisted V2X systems offer advantages over traditional roadside units (RSUs) and purely vehicle-based networks. Their primary strength lies in dynamic adaptability: unlike fixed RSUs, UAVs can be repositioned as needed to provide coverage in response to changing traffic or network conditions. This on-demand deployment reduces the need for ubiquitous physical infrastructure while enabling adaptive network scaling. Studies demonstrate that "flying RSUs" significantly improve connectivity in sparse vehicular networks by filling coverage gaps between distant ground nodes (Hadiwardoyo, 2019)—particularly valuable in rural areas or developing regions where fixed infrastructure deployment is impractical.

From a sensing perspective, drones provide superior vantage points compared to vehicle-mounted sensors alone. Autonomous vehicles' onboard cameras, radar, and LiDAR have limited range and are vulnerable to occlusions from buildings or large vehicles. UAV-mounted sensors mitigate these limitations by observing the environment from above, seeing over obstacles and surveying broader areas simultaneously. This aerial perspective enables more comprehensive situational awareness when integrated with vehicle data.

Communication performance also improves with drone assistance. Signal propagation for V2X radio is often hindered by buildings, terrain, or dense traffic, especially in urban environments. Aerial relays enjoy clearer line-of-sight paths and can maintain simultaneous links with multiple vehicles from elevated positions. By serving as intermediate nodes, drones reduce the number of hops or transmission distances, thereby lowering latency and increasing data rates.

Finally, drone integration can be cost-effective compared to deploying numerous fixed sensors and RSUs. While individual drones represent sophisticated technology investments, their mobility allows them to cover multiple locations over time and be shared among many users as a service. This reduces the need for permanently installed infrastructure that might be underutilized during off-peak hours. In scenarios like temporary events, construction zones, or disaster response, this agility and efficiency far outperform static, traditional infrastructure approaches.

## D.4 TECHNICAL CHALLENGES AND LIMITATIONS

Despite their promise, drone-assisted V2X systems face several technical challenges:

- **Latency and real-time communication:** Supporting safety-critical autonomous driving applications demands ultra-low latency. Introducing drones as relays adds new sources of delay (Gupta & Fernando, 2024). The entire process from capture to broadcast must occur within milliseconds, requiring optimized communication protocols and careful scheduling of V2X message transmissions.

- **Energy constraints:** Limited battery life fundamentally restricts most UAVs to 20-40 minutes of flight time, constraining their endurance for continuous V2X support. Frequent battery swaps or recharging would be required for persistent coverage, while energy budgets also limit onboard sensing and computing capabilities. Energy-efficient hardware and operations (including automated docking stations and solar-powered platforms) remain active research areas.

- **Safety and airspace conflict:** UAVs must avoid collisions with other aircraft and prevent hazards to people and property below. Mid-air collision avoidance requires reliable detect-and-avoid systems, especially at low altitudes around buildings and traffic. Dedicated UAV-to-UAV communication links have been proposed to coordinate movements and prevent incidents. Robust fail-safe protocols (automatic parachutes, controlled emergency landings) are essential to mitigate risks from battery depletion or malfunction.

- **Coordination and scalability:** Managing drone fleets alongside thousands of connected vehicles introduces complex coordination challenges. UAVs must synchronize their trajectories, sensing tasks, and communication resources to maximize coverage without interference. City-wide deployments might require dozens or hundreds of drones, demanding sophisticated aerial traffic management systems. Research continues to explore swarm formation control and adaptive networking algorithms.

- **Security and privacy:** Drone integration expands the attack surface of vehicular networks. Communication links between UAVs and vehicles or infrastructure may be vulnerable to eavesdropping, jamming, or spoofing without proper security measures. Additionally, drone-mounted cameras and sensors may capture sensitive data about individuals or businesses, raising privacy concerns that must be addressed through both regulatory frameworks and privacy-preserving technical designs.

In summary, the integration of unmanned aerial vehicles with vehicle-to-everything communications creates research opportunities driven by both practical needs and technical challenges. While drone-assisted V2X offers compelling advantages in enhanced perception, flexible coverage, and improved communication reliability, it faces substantial hurdles in energy efficiency, latency management, safety, coordination, and security. These challenges establish a rich research landscape spanning communications, sensing, control systems, energy management, and cybersecurity. By addressing these interconnected concerns, researchers can advance drone-assisted V2X from experimental prototypes to practical implementations, ultimately transforming autonomous transportation with dynamic aerial support that overcomes the limitations of traditional ground-based approaches (Wang & Zhang, 2025; Kavas-Torris et al., 2022).

## E  LLM USAGE STATEMENT

Large Language Models (LLMs) were not used to generate, analyze, or create any of the content, results, or figures presented in this paper. LLMs were only employed after the full manuscript was completed, and solely for light editing of grammar and phrasing. All scientific ideas, experimental design, implementation, and writing were conducted entirely by the authors.

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
