# OpenReview forum: "AirV2X: Unified Air-Ground Vehicle-to-Everything Collaboration"
_ICLR.cc/2026/Conference — ICLR 2026 Conference Withdrawn Submission_

### Official Review · Reviewer_W6Xh · 2025-10-25

**Soundness:** 2
**Presentation:** 2
**Contribution:** 3
**Rating:** 4
**Confidence:** 4

**Summary:**

This paper explores a new V2X paradigm for autonomous driving by leveraging vehicle-to-drone (V2D) communication. Whereas prior work primarily studies V2V or vehicle–RSU communication, the authors propose using UAVs as mobile, lower-cost infrastructure that can provide an aerial overview and line-of-sight advantages. To support this study, they introduce a synthetic dataset built with CARLA and AirSim that simulates coordinated vehicle–drone sensing and communication. Experiments compare V2D against conventional V2X setups and indicate that aerial fusion can improve performance (e.g., under occlusions) while potentially reducing infrastructure cost.

**Strengths:**

- `Novel V2X perspective:` Proposes vehicle-to-drone (V2D) collaboration as a cost-effective alternative to fixed RSUs. The aerial viewpoint offers wide FOV and line-of-sight advantages, which plausibly improves perception in occluded scenarios.

- `Reproducible data pipeline:` Provides a clear CARLA+AirSim co-simulation procedure and 6.73 hours of data, enabling the community to study V2D and to recreate/extend the dataset with the described workflow.

- `Empirical support:` 3D object detection experiments indicate that incorporating aerial views can improve perception capability relative to conventional V2X configurations.

**Weaknesses:**

- `Necessity and task coverage:` The benefit of an aerial viewpoint is demonstrated primarily in 3D detection. It is unclear that this advantage transfers to other key perception tasks (online mapping, traffic light recognition, lane/topology extraction, HD map maintenance). Suggest broadening the evaluation to BEV segmentation and lane/centerline extraction.

- `Lack of end-to-end, closed-loop evaluation:`
The paper does not assess whether V2D measurably improves driving quality. Without closed-loop CARLA tests (e.g., Driving Score, route completion, collision/infraction rates), the impact on autonomy is speculative. Recommend integrating aerial fusion into an end-to-end or modular planning stack and reporting improvements under occlusion-heavy routes.

- `Simulation-only evidence and dataset scope:`
The 6.73-hour CARLA+AirSim dataset is useful but synthetic; domain gap to real UAV imagery, weather, lighting, wind-induced motion blur, and sensor noise is unquantified. No real-world validation or transfer experiments are provided.

- `Operational feasibility and total cost of ownership:`
While drones are cheaper per unit than RSUs, recurring costs (battery swaps, maintenance, fleet management, airspace compliance, pilot-in-command requirements) and limited endurance may reduce practical viability. A cost–benefit analysis versus fixed RSUs or hybrid deployments would strengthen the economic argument.

- `Overall assessment:`
The V2D idea is intriguing and potentially impactful, but stronger evidence is needed: broader task coverage, closed-loop driving metrics, realistic comm/latency constraints, and at least preliminary real-world validation or rigorous sim-to-real analysis.

**Questions:**

N/A

---

### Official Review · Reviewer_XxuA · 2025-11-01

**Soundness:** 3
**Presentation:** 4
**Contribution:** 4
**Rating:** 8
**Confidence:** 4

**Summary:**

This paper introduces AirV2X-Perception, a new large-scale, simulated dataset for autonomous driving that focuses on air-ground collaborative perception. The primary contribution is the dataset itself, which, for the first time at this scale, unifies three types of connected agents: ground vehicles, fixed infrastructure (RSUs), and aerial agents (UAVs). Beyond the dataset, the authors provide a comprehensive benchmark of six representative V2X perception algorithms. This benchmark analyzes performance on 3D object detection and BEV semantic segmentation, with a deep dive into robustness against environmental conditions, drone navigation patterns, LiDAR degradation, and spatio-temporal errors.

**Strengths:**

1. The paper's primary contribution is the first large-scale dataset to systematically integrate vehicles, RSUs, and drones, which it clearly positions against prior work. The dataset features impressive scale, agent complexity, and environmental diversity.
2. The paper also provides an extensive and valuable benchmark of perception algorithms. The analysis of drone-specific navigation dynamics (hover, patrol, escort) is a novel and valuable feature.
3. The overall presentation is clear and well-motivated.

**Weaknesses:**

Major Concerns: the sim2real gap. This manifests in two key areas:

1. The specified drone LiDAR configuration (Table 2) is unconventional. A 60-degree vertical FOV pointing exclusively downwards does not correspond to common, commercially available spinning LiDARs. If this setup is purely theoretical, it diminishes the dataset's utility for developing algorithms intended for real-world hardware.
2. While the three drone navigation modes (hover, patrol, escort) are conceptually clear (Section 3.2), the paper omits quantitative flight parameters. Crucial details such as typical/max flight altitude, speed, and the specific constraints for the 'escort' (e.g., following distance, relative altitude) and 'patrol' (e.g., waypoint generation logic) modes are not provided. This lack of detail hinders the assessment of the simulation's realism.
3. Real-world V2X performance is heavily constrained by bandwidth, packet loss, and variable latency. The current benchmark only models a binary "with or without temporal asynchronization", which is not a sufficient proxy for these complex network dynamics. A more realistic benchmark would involve simulating these constraints quantitatively to truly test which algorithms are robust to data loss or high latency.

Minor Concerns:

1. While Table 9 presents 'Sync/Async' and 'Async/Async' results, the paper provides no details on the specific temporal asynchronization parameters (e.g., mean/max latency, distribution of delays) that were simulated, which makes the conclusions about robustness difficult to interpret.

**Questions:**

1. What was the rationale for selecting the 360°H x -30°/-90°V UVA LiDAR configuration? More importantly, is this sensor configuration based on a real-world, commercially available product, or is it a purely theoretical setup?
2. Could the authors provide the specific flight parameters used for the drones, such as the typical altitude for "hover," the altitude and speed ranges for "patrol," and the relative following distance/altitude for "escort"?
3. Given that bandwidth constraints and packet loss are critical challenges in real-world V2X, why were these quantitative network constraints not simulated?
4. For the "Async" experiments in Table 9, what specific inter-agent latency or temporal asynchronization parameters were used? How were these values chosen?

---

### Official Review · Reviewer_4FbW · 2025-11-01

**Soundness:** 3
**Presentation:** 3
**Contribution:** 3
**Rating:** 6
**Confidence:** 3

**Summary:**

This article introduces a UAV dataset for V2X systems, innovatively utilizing bird's-eye-views to address the lack of Road-Side Units in most areas. It achieves a complete simulation dataset integrating multiple scenarios and environments, and designs three types of UAV tasks—Hover, Patrol, and Escort—to adapt to different needs. It supports tasks like 3D object detection and BEV semantic segmentation. Furthermore, it benchmarks existing cooperative perception methods, filling a data gap in aerial-assisted autonomous driving systems.

**Strengths:**

- The dataset includes urban and rural distributions, lighting condition distributions, and weather distributions, and also contains three types of drone missions. Its comprehensive design makes it suitable for robust algorithm evaluation.
- The paper provides a complete evaluation of this dataset for several cutting-edge algorithms, analyzes the performance of various algorithms under different conditions and the reasons behind their performance, and examines the trade-offs between performance and accuracy.
- The paper systematically compares AIRV2X-Perception with multiple V2X datasets, demonstrating its significant advantages and potential in real-world applications.

**Weaknesses:**

- **First, it is worth noting that the author included an arXiv link https://arxiv.org/abs/2506.19283 and a Hugging Face link https://huggingface.co/datasets/xiangbog/AirV2X-Perception/viewer?views%5B%5D=train in the anonymous link. This may violate the anonymity rules, and it need the chairs to decide.**
- The paper mentions that RSUs, due to economic constraints, are only installed at high-traffic intersections and critical urban junctions, while drones, due to their low cost and high dynamic capabilities, can be deployed in various areas. However, in its performance comparison, the article only compared data from three categories: Vehicle + Infra + Drone, Vehicle + Infra, and Vehicle only. This fails to demonstrate the independent performance of the Drone in areas where RSUs are lacking.
- As a simulated dataset, it may not fully cover all real-world scenarios, requiring real-world transfer testing to determine its generalization ability.

**Questions:**

- Could the authors discuss whether some models could be transferred to other datasets (such as OPV2V or V2XSim) for a more intuitive comparison?
- A primary motivation for the paper is that the construction cost of current RSUs is higher than that of drones, but it does not consider the operating costs of drones and the losses incurred in communication and other engineering aspects. Please demonstrate whether the total cost of drones is truly lower than that of RSUs.
- Section 4.4 discusses three drone navigation strategies and concludes that the Patrol mode yields the best results. Is this generalizable? Should different strategies be chosen in different environments and building clusters?

---

### Note · Authors · 2025-11-14

I have read and agree with the venue's withdrawal policy on behalf of myself and my co-authors.